# Assessment of awareness and attitudes of pregnant women toward labor epidural analgesia and factors influencing the decision to opt for it: A cross-sectional study

Adugna Aregawi Kassa[ID]*, Wosenyeleh A. Sahle[1], Tsehaynew Desalew[2], Zufan Lakew[3], Tigist Behabtu[4]

**1** Department of Anaesthesia, Addis Ababa University, Addis Ababa, Ethiopia, **2** Department of Anaesthesia, Hemen Maternal and Child Health Centre, Addis Ababa, Ethiopia, **3** Department of Gynaecology and Obstetrics, Hemen Maternal and Child Health Centre, Addis Ababa, Ethiopia, **4** Department of Midwifery, Hamlin Fistula College of Midwives, Addis Ababa, Ethiopia

* adugna.aregawi@aau.edu.et, adugenet2007@yahoo.com

## Abstract

### Background

Epidural analgesia is one of the most effective forms of pain relief for active first stage and second stages of labor. Despite its many advantages, the technique is not routinely practiced in most centers in developing countries. Awareness and attitudes toward it, as well as the reasons for its limited utilization, have not been explored in our setting. Therefore, the aim of this study was to assess the awareness and attitudes of pregnant women toward labor epidurals and to assess the factors influencing their decision to receive labor epidural analgesia (LEA) during labor and delivery.

### Methods

A cross-sectional study was conducted from June 03 to August 30, 2024 at Hemen Medical Center. A structured questionnaire was used to collect the data. A systematic sampling method was used to collect the data. The collected data were analyzed using SPSS version 26. Descriptive statistics were used to summarize the results, and both bivariate and multivariable logistic regressions were employed to identify factors influencing their decision to receive labor epidural analgesia during labor and delivery. A p-value of < 0.05 was used to determine statistical significance.

### Results

Two hundred thirty pregnant women, with a mean age of 29.9 ± 4.3 years, were included in the analysis. Approximately 78% of the participants had heard of labor epidural analgesia (LEA), although only 6.5% reported healthcare providers as their primary source of information. Approximately 47% believed that LEA was effective

**Data availability statement:** All relevant data are within the paper and Supporting Information files.

**Funding:** The author(s) received no specific funding for this work.

**Competing interests:** The authors have declared that no competing interests exist.

**Abbreviations:** ANC, Antenatal Care; AOR, Adjusted odds ratio; ASA, American Society of Anesthesiologists; CI, confidence interval;-COR, Crude odds ratio; EAA IRERC, Ethiopian Association of Anesthetists Institutional Research Ethics Review Committee; ETB, Ethiopian Birr; HMC, Hemen Medical Center; LEA, Labor epidural analgesia; MCH, Maternal and child health.

for pain relief during labor. However, only 14.3% expressed a willingness to consider using LEA during delivery. Education level, family income, perceived risk to the Fetus, and perceived effectiveness in relieving pain were identified as factors influencing the decision to consider LEA during delivery.

## Conclusion

Despite relatively high basic awareness of LEA, many participants remain hesitant to utilize it during labor and delivery. Comprehensive health education and counselling during antenatal care are essential to address misconceptions and promote informed decision-making regarding LEA.

## Introduction

While childbirth is often anticipated as a joyful milestone, it is also associated with significant pain, which can be one of the most intense experiences in a woman's life [1]. Analgesia during labor and delivery has been approved by the American College of Obstetricians and Gynecologists and the American Society of Anesthesiologists. A maternal request for pain relief during labor has been identified as a sufficient justification [1].

In high-income countries, painless labor is almost universal. However, pain treatment during delivery is still a far-off dream in low-income countries where women are disproportionately burdened by high pregnancy rates and shorter interpregnancy intervals [2].

Epidural analgesia is one of the most effective forms of pain relief during different stages of labor and may allow parturient to rest and relax, facilitating their cooperation during labor and delivery. Despite its benefits, some pregnant women may have reservations about receiving an epidural analgesia because of concerns about potential risks and side effects [3]. Therefore, this study aimed to assess the awareness and attitudes of pregnant women towards labor epidural analgesia (LEA) and assess the factors influencing their decision to consider LEA during delivery.

Labor pain is one of the most intense and challenging experiences faced by women during childbirth, necessitating effective pain management for maternal comfort and positive delivery outcomes [3]. Epidural analgesia is considered the gold standard for labor pain relief because of its effectiveness and high maternal satisfaction [1]. Despite these benefits, the utilization of labor epidural analgesia (LEA) remains low in many developing regions [4,5], including Ethiopia [6]. This underutilization stems from a lack of awareness, misconceptions, and fears regarding the procedure and its potential side effects. Cultural beliefs and misinformation further exacerbate negative attitudes toward LEA [4,5].

There is a scarcity of studies in Ethiopia investigating the awareness and attitudes of pregnant women toward LEA or assessing the factors contributing to its poor utilization. Understanding these factors is crucial for developing educational interventions that could enhance the acceptance and utilization of LEA.

This study aims to fill this critical gap by assessing the awareness and attitudes of pregnant mothers toward labor epidural analgesia and identify the factors influencing their decision-making through a cross-sectional approach. The insights gained will be helpful in developing targeted educational interventions to address misconceptions and fears and integrated into routine antenatal care to ensure that pregnant women receive accurate and comprehensive information about LEA.

The findings will inform training programs for healthcare providers, equipping them with the ability to counsel expectant mothers effectively. This can create a more supportive environment, enhancing the overall childbirth experience. By contributing data from Ethiopia, this study contributes to the global understanding of how cultural, social, and educational factors influence LEA acceptance, aiding in the design of culturally sensitive educational approaches.

Additionally, increasing awareness and positive attitudes toward LEA can improve maternal and neonatal health outcomes by reducing labor-related stress and anxiety. This study's findings will promote effective pain management during labor, thus enhancing maternal and neonatal health in Ethiopia.

Therefore, this facility-based cross-sectional study assessed the level of awareness and attitudes of pregnant mothers toward labor epidural analgesia and to identify factors affecting their decision to consider LEA during delivery at Hemen Medical Center, Addis Ababa, Ethiopia, with the following specific Objectives: to determine the level of awareness of labor epidural analgesia among pregnant mothers, to evaluate the attitudes of pregnant mothers toward labor epidural analgesia, and to identify the factors that influence the decision of pregnant mothers to consider labor epidural analgesia during delivery.

## Materials and methods

### Study area

This study was conducted at HMC. Hemen Medical Center is a private maternal and child care center located in Addis Ababa, Ethiopia. The HMC has been one of the pioneer hospitals in introducing and successfully delivering labor epidural analgesia in the private sector since 2012. On average, approximately 25 labor epidurals are performed each month.

### Study design and period

An institution-based cross-sectional study was conducted from June 03–August 30, 2024.

### Source population

All third-trimester pregnant women attending ANC or delivery services at HMC, Addis Ababa, Ethiopia.

### Study population

All pregnant women in their 3rd trimester of pregnancy (>28 weeks of pregnancy) attended ANC follow-up and planned to have a vaginal delivery at HMC during the study period.

### Inclusion criteria

The inclusion criteria for pregnant women were as follows: they had to be over 18 years old, plan to have a vaginal delivery, be willing to participate in the study and provide informed consent and be in their third trimester of pregnancy. This stage is significant because women in their third trimester are closer to delivery and more likely to have concrete birth plans. During this period, they actively seek and receive information about labor pain management options, including epidural analgesia, which makes their responses more relevant to the study.

### Exclusion criteria

Patients with a previous history of epidural analgesics and pregnant women with a diagnosed fetal anomaly during the current pregnancy were excluded, as psychological and emotional distress may influence their attitudes and decision-making differently from those of the general population. Those who can not read and write were also excluded from the study.

## Sample size determination

Sample size was determined via the single proportion formula and finite population correction formula by assuming P = 0.5, since no similar studies have been conducted in Ethiopia, and a 5% margin of error at the 95% confidence interval was calculated via the following formula:

$$n = \frac{(Z\alpha/2)^2\, P\, (1 - p)}{w^2}$$

where n = sample size, z= 1.96, p= 0.5, w= 0.05, CI= 95% & ἁ= 5%.

$$n = (1.96)2 \times 0.5\, (1 - 0.5)/ (0.05)2 = 384$$

The final sample size, nf = n/(1 + n/N), where N = the estimated number of 3rd trimester pregnant women expected to visit HMC for ANC follow-up within the study period, 2 months, is approximately 880.

Therefore, a correction formula was applied and the final sample size: nf = 384/ (1 + 384/880) = **267**. A total sample of 267 3rd trimester pregnant women were included in the study within a 2-month period.

## Sampling technique

A systematic sampling technique was used. During the data collection period, 880 third-trimester pregnant women were expected to visit the ANC at HMC within three months. Therefore, one out of every three eligible subjects was selected from the ANC appointment lists.

## Dependent variables

Awareness and attitudes of mothers toward labor epidural analgesia and the decision to consider LEA during delivery.

## Independent variables

Maternal age, parity, educational level, income, awareness of epidural analgesia, attitude toward LEA, source of information about labor epidural, and previous childbirth experience.

## Data collection process and technique

Data were collected via a structured, self-administered questionnaire. The questionnaire was developed by adapting the validated 'Labor Pain Relief Attitude Questionnaire for pregnant women (LPRAQ-p) [7]. After contextualizing it to our setting through literature review, we translated the instrument into Amharic, the questionnaire was distributed to pregnant mothers during their antenatal visits at the hospital. Eligible mothers were identified from the daily registry and approached by the triage nurse. After explaining the study's purpose, willing participants received an information sheet and consent form (detailing voluntary participation). Upon signing, the nurse clarified questionnaire instructions and addressed queries. Participants completed the 15–20-minute survey prepared in Amharic language in the OPD waiting area and returned it to the nurse once completed.

The questionnaire assessed their level of awareness and attitudes toward labor epidural analgesia, whether they planned to opt for it during delivery, and the factors influencing their decision to choose LEA.

## Data quality assurance

To ensure the quality of the data, training on the objectives and relevance of the study and brief orientations on the assessment tools were provided for the data collectors. After data collection, each questionnaire was checked by the investigator for completeness.

### Data analysis

Completed data were analyzed via IBM SPSS Statistics for Windows, version 26 (IBM Corp., Armonk, NY, USA). Descriptive statistics, such as frequencies and percentages, were used to summarize the questionnaire responses.

Prior to conducting multivariable logistic regression, multicollinearity among the independent variables was assessed using the Variance Inflation Factor (VIF) to detect potential collinearity. All variables had VIF values less than 10, indicating no significant multicollinearity. Following this, bivariate logistic regression was performed for each independent variable in relation to the dependent variable. Variables with a p-value less than 0.2 were selected for inclusion in the multivariable logistic regression model. In the final model, variables with a p-value less than 0.05 were considered statistically significant independent predictors.

### Ethical considerations

This study followed the Helsinki protocols for human research strictly. The study protocol was reviewed and approved by the Ethiopian Association of Anesthetists Institutional Research Ethics Review Committee (EAA IRERC) (Protocol Number: EAA/16/30895/005). Data were collected using self-administered questionnaires from pregnant women in their third trimester who were attending antenatal care (ANC) follow-up at Hemen Medical Center between June 3, 2024, and August 30, 2024. A written informed consent was obtained from participants, and those who were willing to take part in the study confirmed their consent by signing the consent form provided with the questionnaire. They were assured of confidentiality and informed that their participation was entirely voluntary. To maintain privacy, the completed questionnaires were securely stored in a locked location.

## Results

### Demographic characteristics of the participants

A total of 267 pregnant women attending antenatal care (ANC) follow-up at Hemen Medical Centre participated in the study. Of these, 37 either did not return the questionnaire or submitted incomplete responses, resulting in a response rate of 86%. Therefore, 230 participants were included in the analysis. The mean age of the participants was 29.9 ± 4.3 years, and approximately 70% of the participants were in the 25--34 years age group. The majority of the respondents (66.6%) were degree holders or higher, and approximately 75% were employed in various occupations. The monthly family income varied, with approximately 20% earning less than 10,000 ETB, 11% reporting an income above 50,000 ETB. In terms of obstetric history, the mean gestational age of the participants was 37.8 ± 1.8 weeks, and approximately half of the participants were nulliparous (Table 1).

### Awareness of labor epidural analgesia

Among the total participants, 78% reported having heard of labor epidural analgesia (LEA). The primary sources of information were previous users (27%), followed by family and friends (24%) and the internet (19.6%). Only 6.5% of the respondents cited healthcare providers as their primary source, underscoring the need for direct communication to enhance awareness. Additionally, only 34% of the participants provided explanations from health care providers about LEA during their ANC follow-up. Moreover, a significant portion of the participants (21.7%) had not previously heard of LEA.

### Attitudes Toward Labor Epidural Analgesia

Regarding attitudes, 47% of women believed that LEA was effective for pain relief during labor. However, only 14.3% expressed a willingness to consider using LEA during delivery, while the rest did not have a plan at all (47.8%) or were undecided to take LEA (37.8%). The main concerns among those unwilling or hesitant to use LEA included fear of side

**Table 1. Sociodemographic and obstetric characteristics of third trimester pregnant women attending ANC follow-up at HMC, Ethiopia.**

| Variables | Category | Frequency(n) | Percentage |
|---|---|---|---|
| Maternal age group | 18-25 years old | 22 | 13.2% |
| | 26-34 years old | 117 | 70% |
| | ≥35 years old | 28 | 16.8% |
| Level of Education | Primary School | 9 | 4% |
| | Secondary School | 27 | 11.7% |
| | Diploma | 41 | 17.8% |
| | Degree | 100 | 43.5% |
| | Masters and above | 53 | 23% |
| Occupation | Unemployed | 7 | 3% |
| | Stay at home mother | 51 | 22.2% |
| | Government Employee | 61 | 26.5% |
| | Private Employee | 70 | 30.4% |
| | Private business owner | 41 | 17.8% |
| Monthly family income | <10, 000 ETB | 46 | 20% |
| | 10,000–25,000 ETB ($100 - $250) | 84 | 36.5% |
| | 25,000–50,000 ETB ($250 - $500) | 74 | 32.2% |
| | >50, 000 ETB (>$500) | 26 | 11.3% |
| Parity | Nulliparous | 112 | 48.7% |
| | Primiparous | 66 | 28.7% |
| | Multiparous | 52 | 22.6% |

effects for the mother (20%), concerns for the baby's safety (11%), interference with the natural labor process (11%), and additional cost (20%).

**Factors influencing decision-making regarding LEA during laboring**

Bivariate logistic regression analysis revealed that education level, occupation, family monthly income, perception of LEA causing significant problems for the mother, perceived problem for the Fetus, and awareness of its pain-relief effectiveness were factors associated with the decision to receive LEA, with a p value < 0.2. These variables were included in the multivariable logistic regression analysis.

The multivariable logistic regression analysis revealed that education level, family monthly income, the perception that LEA could cause significant problems for the Fetus, and the perception of LEA effectiveness in relieving pain were significantly associated with the decision to receive LEA during labor and delivery, with a p value < 0.05 (Table 2).

## Discussion

This study revealed that 78% of the participants had prior awareness of labor epidural analgesia (LEA). Among those who were aware, only 6.5% identified healthcare providers as their primary source of information. Furthermore, only 14.3% of the participants expressed a willingness to receive LEA during delivery. The main reasons for hesitancy or unwillingness to accept LEA included concerns about potential side effects for the baby, doubts about its pain relief effectiveness, lower family income, and lower educational level.

**Table 2. Factors influencing the decision to receive LEA during labor of participants at HMC, Ethiopia, 2024.**

| Variables | | Decision to take LEA | | COR (95% CI) | AOR (95% CI) | P value |
|---|---|---|---|---|---|---|
| | | Yes | No/Not sure | | | |
| Education | Diploma and Lower | 3 (4.4%) | 66 (95.6%) | 1 | 1 | **0.021*** |
| | Degree and above | 30 (18.6%) | 131 (81.4%) | 5.01 (1.48, 17.1) | 1.9 (1.28, 6.78) | |
| Occupation | Unemployed | 5 (8.6%) | 53 (91.4%) | 1 | 1 | 0.402 |
| | Employed | 28 (16.3%) | 144 (83.7%) | 2.1 (1.06, 5.6) | 1.6 (0.53, 4.8) | |
| Income | <25,000 ETB | 12 (9%) | 109 (91%) | 0.31 (0.3, 0.79) | 0.28 (0.11, 0.69) | **0.006*** |
| | ≥25,000 ETB | 21 (19.3%) | 88 (80.7%) | 1 | 1 | |
| Perceived problem on mother | No | 17 (22.6%) | 58 (77.4%) | 1 | 1 | 0.65 |
| | Yes/Not sure | 16 (10.3%) | 139 (89.7%) | 0.4 (0.19 – 0.83) | 0.82 (0.34, 1.96) | |
| Perceived problem on Fetus | No | 23 (29.1%) | 56 (70.9%) | 1 | 1 | **<0.001*** |
| | Yes/Not sure | 10 (6.6%) | 141 (93.4%) | 0.17 (0.08 – 0.39) | 0.18 (0.07, 0.45) | |
| Pain relief perception | Yes | 27 (25%) | 81 (75%) | 1 | 1 | **0.001*** |
| | No/not sure | 6 (4.9%) | 116 (95.1%) | 0.16 (0.06, 0.4) | 0.18 (0.06, 0.49) | |

Note: LEA = labor epidural analgesia, COR = crude odds ratio, AOR = adjusted odds ratio, CI = confidence interval

*Statistically significant association at P < 0.05

The relatively high level of awareness observed in this study may be attributed to nearly all participants (98%) being from urban areas, specifically Addis Ababa, the capital city of Ethiopia. Urban residency likely increases exposure to advanced obstetric practices, either directly or indirectly [8].

Our findings are consistent with studies conducted in urban settings. For example, a study among women residing in Riyadh reported that most participants demonstrated good knowledge of epidural analgesia for labor pain relief [9]. Similarly, another study conducted in the same region also documented adequate knowledge among participants regarding epidural analgesia [10].

In contrast, numerous studies conducted in developing countries have reported low levels of awareness of labor analgesia. For example, two Ethiopian studies highlighted that the overall awareness of labor analgesia among pregnant women was low [8,11]. A study conducted in Nigeria also reported low awareness levels of labor epidural analgesia among its participants [12]. Similarly, studies from India identified a lack of awareness about labor epidural analgesia in the majority of their study populations [3,13].

Our study revealed that only 14.3% of participants were willing to consider labor epidural analgesia (LEA) during delivery, despite 47% acknowledging its effectiveness in relieving labor pain. The primary reasons for their reluctance included concerns about potential side effects for the Fetus. However, these perceptions are largely unfounded, as recent studies have demonstrated the safety of LEA for both maternal and fetal outcomes [14–16].

Our study revealed that educational level, monthly family income, concern regarding the potential adverse effects of LEA on fetal well-being, and perceived efficacy of LEA in pain relief were factors influencing the decision to perform LEA during labor and delivery.

Compared with those with a diploma or lower educational attainment, those with a degree or higher education were nearly twice as likely to opt for LEA (P = 0.021, AOR: 1.9). In line with our findings, a study conducted in Ethiopia reported that a higher education level is associated with a positive attitude toward labor analgesia [11]. Multiple other studies conducted in different parts of the world have also identified the level of education as an important factor influencing pregnant women's decision to request LEA [9,17–19]. This association may be explained by the fact that individuals with higher education typically have greater access to reliable health information and are more likely to engage in effective

communication with healthcare providers. These factors contribute to an enhanced understanding and acceptance of LEA as an evidence-based option for pain management during labor [3,18,19].

In this study, monthly family income was identified as a factor influencing women's decision to utilize labor epidural analgesia (LEA). According to our findings, women with a monthly family income exceeding 25,000 ETB ($250) were 3.5 times more likely to seek LEA during labor and delivery than were those with a lower income (P = 0.006, AOR: 0.28). Multiple studies have similarly reported that income is significantly associated with the decision to opt for LEA [9,18,20]. A possible explanation for this finding is that LEA adds to the overall cost of care. The additional expense associated with the technique was frequently mentioned by our study participants as a reason for their unwillingness to choose LEA.

Perceived concerns about potential harm to the Fetus emerged as a significant factor influencing the willingness to accept LEA in our study. The participants who believed that LEA caused significant harm to their baby were more than five times more likely to refuse LEA during labor and delivery (P < 0.001, AOR = 0.18). Consistent with our findings, previous studies have reported similar observations [3,21]. However, substantial evidence indicates that LEA does not pose any risk to the Fetus [15,16,22]; in contrast, it may confer benefits, such as improved oxygenation and higher APGAR scores at delivery [14,23]. These results highlight how misconceptions and a lack of health education can negatively influence decisions regarding quality healthcare. The evidence suggests that health education can play a critical role in improving women's decisions to opt for LEA during labor and delivery [24,25].

Our study also revealed that pregnant women who believe that LEA is effective in preventing labor pain are more than six times more likely to choose LEA during delivery (P = 0.001, AOR = 0.18). Similarly, other studies have reported that when pregnant women perceive LEA as effective in relieving labor pain, the likelihood of choosing it significantly increases [13,24–26]. These findings underscore the importance of educating women about the effectiveness of LEA in managing labor pain during antenatal care follow-up visits.

## Strengths and limitations of the study

### Strength of the study

This study is the first in Ethiopia to systematically examine pregnant women's awareness and attitudes toward labor epidural analgesia (LEA), addressing a critical gap in national labor analgesia research. Its strength lies in the use of a structured, culturally adapted questionnaire and systematic sampling, which improves the reliability of the findings. By generating locally relevant evidence, the study provides a foundation for targeted educational interventions and policies to support informed decision-making and improve labor pain management.

### Limitations

Our study has some limitations. First, we were unable to assess the impact of previous epidural experience on maternal decisions regarding labor epidural analgesia (LEA), as LEA is a relatively new practice in our country and few women have prior exposure to it.

Second, the study was conducted at a private Maternal and Child Health (MCH) center located in the capital city, where participants attending there generally have greater awareness than the general population. This may limit the generalizability of the findings to other obstetric populations in Ethiopia. Therefore, future research conducted in multi-center hospitals representing both urban and rural populations would be highly recommended. Lastly, the cross-sectional design of the study precluded us from tracking participants to determine how many ultimately opted for LEA during delivery.

## Conclusions

This study highlights that while basic awareness of LEA is relatively high, many participants remain hesitant to utilize it during labor and delivery. This reluctance appears to stem from inadequate explanations provided by healthcare professionals, potentially leading to negative perceptions. The factors significantly associated with participants' decisions to seek

LEA included educational level, income, perceived risks to the Fetus, and perceptions of pain relief effectiveness. These findings underscore the need for comprehensive health education and counselling during antenatal care. To improve LEA uptake, healthcare providers should deliver structured antenatal counselling on risks and benefits, while administrators must train staff and provide educational materials to ensure consistent messaging.

## Supporting information

**S1 File. SPSS dataset–Awareness and factors influencing decision making for labor epidural analgesia.**
(SAV)

## Acknowledgments

We extend our gratitude to the Hemen Maternal and Child Health Centre and its staff for their invaluable support in facilitating this study. We also deeply appreciate the participating women for sharing their experiences, which were instrumental to this research.

## Author contributions

**Conceptualization:** Adugna Aregawi Kassa, Tsehaynew Desalew, Zufan Lakew, Tigist Behabtu.

**Data curation:** Wosenyeleh A. Sahle.

**Formal analysis:** Adugna Aregawi Kassa.

**Funding acquisition:** Zufan Lakew.

**Methodology:** Adugna Aregawi Kassa, Wosenyeleh A. Sahle, Tsehaynew Desalew, Tigist Behabtu.

**Project administration:** Zufan Lakew.

**Resources:** Tsehaynew Desalew.

**Supervision:** Wosenyeleh A. Sahle, Tsehaynew Desalew, Zufan Lakew, Tigist Behabtu.

**Validation:** Adugna Aregawi Kassa, Tigist Behabtu.

**Writing – original draft:** Adugna Aregawi Kassa.

**Writing – review & editing:** Wosenyeleh A. Sahle, Zufan Lakew, Tigist Behabtu.

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
