## [Decision Letter · Decision Letter 0]

8 Apr 2025

PONE-D-24-60257Assessment of awareness and attitudes of pregnant women toward labor epidural analgesia and factors influencing the decision to opt for it: A cross-sectional study.PLOS ONE

Dear Dr. Kassa,

Thank you for submitting your manuscript to PLOS ONE. After careful consideration, we feel that it has merit but does not fully meet PLOS ONE’s publication criteria as it currently stands. Therefore, we invite you to submit a revised version of the manuscript that addresses the points raised during the review process.

We look forward to receiving your revised manuscript.

Kind regards,

Dereje Zewdu Assefa, BSc, MSc

Academic Editor

PLOS ONE

**Journal Requirements:**

Please ensure that your manuscript meets PLOS ONE's style requirements, including those for file naming. The PLOS ONE style templates can be found at https://journals.plos.org/plosone/s/file?id=wjVg/PLOSOne_formatting_sample_main_body.pdf and https://journals.plos.org/plosone/s/file?id=ba62/PLOSOne_formatting_sample_title_authors_affiliations.pdf 2. Please provide additional details regarding participant consent. In the ethics statement in the Methods and online submission information, please ensure that you have specified (a) whether consent was informed and (b) what type you obtained (for instance, written or verbal, and if verbal, how it was documented and witnessed). If your study included minors, state whether you obtained consent from parents or guardians. If the need for consent was waived by the ethics committee, please include this information. If you are reporting a retrospective study of medical records or archived samples, please ensure that you have discussed whether all data were fully anonymized before you accessed them and/or whether the IRB or ethics committee waived the requirement for informed consent. If patients provided informed written consent to have data from their medical records used in research, please include this information. 3. In the online submission form, you indicated that Data can be made available to researchers upon request by contacting the authors.  All PLOS journals now require all data underlying the findings described in their manuscript to be freely available to other researchers, either a. In a public repository, b. Within the manuscript itself, or c. Uploaded as supplementary information.This policy applies to all data except where public deposition would breach compliance with the protocol approved by your research ethics board. If your data cannot be made publicly available for ethical or legal reasons (e.g., public availability would compromise patient privacy), please explain your reasons on resubmission and your exemption request will be escalated for approval.

**Additional Editor Comments:**

In the abstract's background section, the authors state that “Epidural analgesia is one of the most effective forms of pain relief during various stages of labor.” Since epidural labor analgesia is only administered during the first and second stages of labor, it would be clearer to specify "first and second stages of labor" instead.

Please ensure that the main aim of the study aligns consistently with the outcomes. The abstract's background section states that the study aims to assess pregnant women's awareness and attitudes toward labor epidurals, as well as the factors influencing their decision to receive labor epidural analgesia (LEA) during labor and delivery. However, the methods section of the abstract mentions the goal of identifying factors associated with the decision to receive labor epidural analgesia during delivery.

In the abstract section, please clearly mention that you used multivariable logistic regression analysis instead of logistic regression.

In the introduction section, please remove the terms "background," "statement of the problem," and "significance of the study." Additionally, ensure your manuscript adheres to the journal's guidelines.

Please improve the grammar and typos throughout the manuscript.

The mean age of the participants mentioned in the abstract differs from that in the results section. Please correct this discrepancy. Additionally, there is no need to compare USD to Ethiopian birr, as currency values fluctuate over time in the result section.

Since the calculated sample size and the number of included participants differ, please include the response rate in the Results section.

In the data analysis section, you did not address multicollinearity, explain how bivariate analysis was conducted, or detail how variables were selected for multivariable logistic regression from the bivariate analysis. Please provide a clear description of the data analysis techniques used.

In the discussion section, please omit the written subgroup and begin with your main findings.

In line 216, the authors referred to the subgroup title as factors influencing the decision to perform LEA. This should be revised to reflect that it is the decision to receive or accept LEA, as pregnant women do not have the authority to perform the procedure themselves.

In the limitations section, it is important to note that extrapolating these findings to the entire obstetric population is challenging, as the study was conducted in private maternal and child care center in the capital city, where pregnant women tend to have better awareness.

In the limitations section, it is important to note that extrapolating these findings to the entire obstetric population is challenging. This is because the study was conducted in a private maternal and child care center in the capital city, where pregnant women typically have better awareness. Future research conducted in multicentre hospitals that represent both urban and rural populations would be highly beneficial.

Reviewers' comments:

Reviewer's Responses to Questions

**Comments to the Author**

1. Is the manuscript technically sound, and do the data support the conclusions?

Reviewer #1: Yes

2. Has the statistical analysis been performed appropriately and rigorously? 

Reviewer #1: Yes

3. Have the authors made all data underlying the findings in their manuscript fully available?

Reviewer #1: Yes

4. Is the manuscript presented in an intelligible fashion and written in standard English?

Reviewer #1: Yes

5. Review Comments to the Author

**Reviewer #1: ** Dear authors,

Thank you for submitting this interesting paper investigating the awareness and attitudes of pregnant women toward labor epidural analgesia and factors influencing their decision making. While labor epidural analgesia is a well-accepted form of pain relief in many high-income countries, in low-middle income countries, its awareness, availability, and use are often limited due to a combination of economic, cultural, healthcare system, and educational barriers. Addressing these challenges requires multifaceted approaches, such as improving access to healthcare, raising awareness about pain management options, training more healthcare providers, and addressing economic constraints, to empower women in making informed decisions about their labor experiences.

The manuscript is technically sound, and the methods permitted the statistical analysis of the data. Data that supported the conclusions.

This article could be used as an advocacy, showing health authorities that addressing the barriers from multiple angles (education, access, training, and infrastructure) could improve the awareness, acceptance, and use of labor epidural analgesia in low-middle income countries, ultimately contributing to better maternal health outcomes and improving the childbirth experience for women.

Thank you for your work.

6. PLOS authors have the option to publish the peer review history of their article (what does this mean? ). If published, this will include your full peer review and any attached files.

**Do you want your identity to be public for this peer review?** For information about this choice, including consent withdrawal, please see our Privacy Policy .

Reviewer #1: No

---

## [Author Response · Author response to Decision Letter 1]

22 May 2025

The manuscript revised according to PLOS ONE formatting style and the following changes are made: The title, author names, affiliations, and the main body were revised according to the PLOS ONE formatting requirements.

Also, all additional comments from the editor and reviewers have been explicitly reviewed and incorporated in to the revised manuscript.

---

## [Decision Letter · Decision Letter 1]

27 Jul 2025

PONE-D-24-60257R1Assessment of awareness and attitudes of pregnant women toward labor epidural analgesia and factors influencing the decision to opt for it: A cross-sectional study.PLOS ONE

Dear Dr. Kassa,

Thank you for submitting your manuscript to PLOS ONE. After careful consideration, we feel that it has merit but does not fully meet PLOS ONE’s publication criteria as it currently stands. Therefore, we invite you to submit a revised version of the manuscript that addresses the points raised during the review process.

We look forward to receiving your revised manuscript.

Kind regards,

Dereje Zewdu Assefa, BSc, MSc

Academic Editor

PLOS ONE

Journal Requirements:

Additional Editor Comments:

I believe the manuscript has significantly improved since the first submission. However, based on the new reviewers' comments and my careful review, I still recommend major revisions. Please provide a point-by-point response in a separate document to save time for the reviewers and editors.

Reviewers' comments:

Reviewer's Responses to Questions

**Comments to the Author**

1. If the authors have adequately addressed your comments raised in a previous round of review and you feel that this manuscript is now acceptable for publication, you may indicate that here to bypass the “Comments to the Author” section, enter your conflict of interest statement in the “Confidential to Editor” section, and submit your "Accept" recommendation.

Reviewer #2: (No Response)

Reviewer #3: All comments have been addressed

Reviewer #4: (No Response)

2. Is the manuscript technically sound, and do the data support the conclusions?

Reviewer #2: Partly

Reviewer #3: Yes

Reviewer #4: Yes

3. Has the statistical analysis been performed appropriately and rigorously? 

Reviewer #2: Yes

Reviewer #3: Yes

Reviewer #4: Yes

4. Have the authors made all data underlying the findings in their manuscript fully available?

Reviewer #2: Yes

Reviewer #3: Yes

Reviewer #4: Yes

5. Is the manuscript presented in an intelligible fashion and written in standard English?

Reviewer #2: No

Reviewer #3: Yes

Reviewer #4: Yes

6. Review Comments to the Author

Reviewer #2: General comment

This cross-sectional study aims to evaluate the awareness and attitudes of pregnant women regarding labor epidural analgesia, as well as the factors that influence their decision to choose this option (Manuscript Number PONE-D-24-60257).

While the study's objective is clearly articulated and holds significant importance, the introduction, methods, and discussion sections require substantial enhancement.

It is advisable for the authors to consult the STROBE guidelines for cross-sectional studies to ensure comprehensive reporting and to explicitly state adherence to these guidelines within the manuscript.

Additionally, a thorough revision of the English language and proofreading is recommended, along with a major revision of the manuscript itself.

In the abstract

It is essential to maintain proper spacing between text and punctuation throughout the manuscript. For example, in line 22, there should be a space after "countries." and before "Awareness," and in line 23, a space is needed before "Therefore."

Furthermore, in line 25, the authors should specify the exact time frame of the study conducted from June to August 2024.

It is also important to clarify the sampling method used

The authors should indicate that as the study was conducted in the facility. Like a facility-based cross-sectional study design.

In line 28, the phrase "multivariable logistic regression" should be revised to "Both bivariable and multivariable logistic regressions" to accurately reflect the analysis performed.

Lastly, in line 20, the statement regarding epidural analgesia should be modified to specify "active first stage and second stage of labor," as the intensity of pain increases with cervical dilation beyond 4 cm, and the analgesia is most effective during this phase. It is crucial to note that administering epidural analgesia during the latent phase of labor may hinder labor progression.

Keywords: Labor Epidural Analgesia, Awareness and Attitudes, Decision-Making Factors, Pregnant Women.

Please separate the awareness from attitude by using comma, and remove the word ‘and’

Introduction

The introduction needs to be more focused on what the problem is

The authors fail to clarify their criteria for selecting the literature they reference, which diminishes the credibility of their review.

Furthermore, the literature review is notably limited, primarily drawing from studies within a single geographical region, rather than considering lessons from other parts of Africa or globally.

It would enhance the introduction to include a discussion of various labor pain relief methods, both pharmacological and non-pharmacological, along with evidence supporting their effectiveness and implications for birth outcomes.

Additionally, there is a lack of reference to the content in lines 61-63, which discusses the underutilization of labor epidural analgesia due to misconceptions and cultural beliefs that foster negative attitudes.

The citation in reference number 6, mentioned in line 61, does not pertain to epidural analgesia and should be verified by the authors.

Lastly, the introduction suffers from redundancy, particularly in the text found in lines 55-56, which could be streamlined for clarity.

The methodology section of this study requires further elaboration, particularly regarding the specific procedures employed and the population densities in the areas under investigation.

Additionally, the authors have not disclosed the number of anesthesiologists involved in the study, nor have they clarified whether the use of epidural analgesia during labor incurs any costs at their facility.

There is also a lack of information regarding the total number of deliveries performed at the health facility and the qualifications of the personnel involved.

Furthermore, the staff-to-patient ratios remain unspecified, and there is no mention of the training that providers receive, either pre-service or in-service, concerning epidural analgesia in labor.

It is essential for the authors to distinguish between the source population and the study population, as well as to justify the exclusion of pregnant women in their first and second trimesters.

The rationale behind the use of a correction formula for sample size calculation should also be addressed.

Lastly, the dependent variables should be revised to focus solely on the "Awareness and attitudes of mothers toward labor epidural analgesia," omitting the reference to the decision to consider LEA during delivery.

The authors did not provide details regarding their approach to engaging study participants, including the methods of invitation, the locations where interviews were conducted, and the average duration of these interviews for completing the questionnaire.

It is essential to clarify the original source of the questionnaire, including the context in which it was first utilized, and whether it underwent validity and reliability testing at that time.

Additionally, if any modifications were made to the tool, an explanation for these changes should be included.

The questionnaire used should be described, referred to within the manuscript, and made available as an appendix

The methods section ought to outline the structure of the tool, detailing the number of sections, the content and purpose of each section, and the number of items or questions per section. This information will enhance the reader's understanding of how the tool facilitated the researchers in addressing the research questions and assessing outcomes.

Furthermore, incorporating a conceptual framework that illustrates the anticipated relationships among the study variables would provide clarity on the variables adjusted for in the multivariable logistic regression model.

Lastly, it is important to describe the background of the data collectors, such as their qualifications as health professionals.

Result

In table 1, the criteria used to classify study participants into income brackets, such as <10,000 ETB and 10,000-25,000 ETB, should be clearly defined.

In Table 2, the phrase "Decision to take LEA (Yes/No/Not sure)" lacks clarity and requires further explanation to ensure that readers fully understand the context and implications of these responses.

Discussion

This section requires significant improvement to effectively illustrate the connections among various components of the research paper and to highlight the significance of the findings.

The authors have not sufficiently analyzed, explained, or interpreted their results; instead, they have largely reiterated the content from the results section and merely compared their findings to previous studies.

A concise summary of the key findings should be presented at the beginning, contextualized within the existing literature.

Furthermore, the recommendations appear more suited for a report than a scientific publication, often lacking clarity due to missing words.

The authors fail to specify who should implement the suggested actions or how these actions would be executed.

Limitations

The exclusion of mothers in labor represents a notable limitation of this study, as it may significantly affect the assessment of the need for labor pain relief.

Thank you!

Reviewer #3: This is a good piece of evidence. The manuscript is well-written and important points are being mentioned.

Reviewer #4: Dear Editor,

I appreciate the invitation to discuss the intriguing title, "Assessment of Awareness and Attitudes of Pregnant Women toward Labor Epidural Analgesia and Factors Influencing the Decision to opt for it: A Cross-Sectional Study." This study promises to offer valuable insights for enhancing obstetric care, management, and outcomes.

The manuscript is well-written and easy to understand. I appreciate the authors for their effort in producing this work.

I have some comments and suggestions, which you can find below.

General suggestions

- Adhere with the journal guidelines.

- Please revise the introduction section, as it appears fragmented. Additionally, some sentences lack references. Acknowledge the sources, or if they are from similar sources, use conjunctions to combine them. Some paragraphs feel disjointed. Use transitional phrases like “However,” “In contrast,” or “Furthermore” to create flow.

- Try to address punctuation errors throughout the document.

- Make the references uniform example reference 17

Introduction section: -

-Avoid generalizations such as "worst pain ever." Therefore, modify lines 44-45 to read: "While childbirth is often anticipated as a joyful milestone, it is also associated with significant pain, which can be one of the most intense experiences in a woman’s life."

- I do not see the importance of the sentence mentioned by the authors in lines 52-54, particularly regarding your study's aim. It would be better for the authors to omit it.

- The authors stated in lines 64-65 that "there are no published studies in Ethiopia investigating the awareness and attitudes of pregnant women toward LEA or assessing the factors contributing to its poor utilization." However, reference 10, conducted by Workie MM et al. in 2021, presents a similar study. Therefore, please revise this justification section.

- I recommend that the authors revise the justification of the study, as it lacks coherence and contains redundant phrases, such as "the aim of this study," in multiple paragraphs (e.g., lines 68-70 and 81-83). Additionally, it should be made more concise while maintaining the flow of ideas and improving the overall presentation.

Method section

-The authors should format the inclusion criteria as a heading to ensure consistency with the other headings.

- Why did the authors exclude patients with a previous history of epidural analgesics? I recommend that they consider this factor when determining the outcome. The reasons included in lines 101-102 do not seem particularly necessary. Moreover, if the questionnaire was self-administered, please note that individuals who cannot read or are not educated were excluded.

- Sample size determination section: If there were 880 individuals expected to attend ANC follow-up, why did the authors not use the calculated sample size of 384 instead of applying the population reduction formula?

- Sampling technique: Was the list of 880 individuals accessible for systematic random sampling? I am concerned that not all expected individuals may attend the full ANC follow-up during the third trimester of pregnancy for various reasons.

-Data analysis: The authors used the phrase "bivariate analysis" in lines 131 and 133, which is not appropriate in the context of this study. This term implies a relationship between two predictors, rather than between a predictor and an outcome. Therefore, it should be changed to "bivariable analysis." Please address similar errors throughout the manuscript example there is similar error in line number 174.

- Ethical considerations: The authors need to include the fact that the study followed the Helsinki guidelines for human research in this section.

Result section

-In line 149, please insert a full stop before the word “Therefore.” Additionally, in line 151, omit the use of the double hyphen in “25--34 years age group.”

- In line 175, the phrase “Perception of LEA causing significant problems for the mother and fetus” needs modification, as the perceived problems for the mother and fetus are distinct contributing factors. Please split it into two separate statements.

-In the tables, please ensure they are self-explanatory regarding why, where, and when. The authors missed the year of study.

- In "Table 2," under the variable "Education," the authors listed the p-value for the variable they considered constant. This may be a typographical error, so please correct it. Additionally, please remove the merging of the table in the boxes indicating the p-value to reduce ambiguity for readers.

Discussion section

- The authors were certain that the causes of the unwillingness to accept LEA include perceived interference with the labor process and the additional costs associated with the technique. However, these factors were not studied, so where did you obtain these reasons?

- In line 194, the authors mention “the relatively high level of awareness observed in this.” What does this relatively high awareness mean? Compared to what? Is there any standard for this assessment?

- Overall, in the discussion section, the authors should refrain from hastily generalizing their findings in relation to previous studies, such as those from urban research. In low-income countries, where awareness levels are often low, it is essential to include specific numeric figures, such as the percentage of awareness, along with appropriate citations for each statistic. Additionally, they should ensure that the percentages from previous studies fall within the confidence interval of the current study's outcome values to say consistent with the current study. Similarly, in lines 218-220, the authors state that “Multiple other studies conducted in different parts of the world have also identified the level of education as an important factor influencing pregnant women’s decision to request LEA.” Please specify the countries where these studies were conducted, along with the respective citations, rather than citing them collectively. This suggestion also applies to other contributing factors.

Conclusion section

- The authors conclude that in lines 259-260, “This reluctance appears to stem from inadequate explanations provided by healthcare professionals, potentially leading to negative perceptions.” However, these variables were not associated with your outcome, so please base your conclusions on your findings.

7. PLOS authors have the option to publish the peer review history of their article (what does this mean? ). If published, this will include your full peer review and any attached files.

**Do you want your identity to be public for this peer review?** For information about this choice, including consent withdrawal, please see our Privacy Policy .

Reviewer #2: No

Reviewer #3: **Yes: ** Tabeer Tanwir Awan

Reviewer #4: **Yes: ** Temesgen Birlie Asmare

---

## [Author Response · Author response to Decision Letter 2]

5 Aug 2025

All reviewer comments have been addressed and are attached as a point-by-point response table.

---

## [Decision Letter · Decision Letter 2]

13 Oct 2025

PONE-D-24-60257R2Assessment of awareness and attitudes of pregnant women toward labor epidural analgesia and factors influencing the decision to opt for it: A cross-sectional study.PLOS ONE

Dear Dr. Kassa,

Thank you for submitting your manuscript to PLOS ONE. After careful consideration, we feel that it has merit but does not fully meet PLOS ONE’s publication criteria as it currently stands. Therefore, we invite you to submit a revised version of the manuscript that addresses the points raised during the review process.

We look forward to receiving your revised manuscript.

Kind regards,

Dereje Zewdu Assefa, BSc, MSc

Academic Editor

PLOS ONE

Journal Requirements:

Additional Editor Comments:

The abstract need to be presented with subtitle of background, methods, results and conclusion per the journal standard.

Please avoid bullet point at the end of introduction.

The document still needs language improvement, typos and grammatical correction throughout the entire document.

Please revise the inclusion criteria as follows; “The inclusion criteria for pregnant women were as follows: they had to be over 18 years old, plan to have a vaginal delivery, be willing to participate in the study and provide informed consent, and be in their third trimester of pregnancy. This stage is significant because women in their third trimester are closer to delivery and more likely to have concrete birth plans. During this period, they actively seek and receive information about labor pain management options, including epidural analgesia, which makes their responses more relevant to the study.”

Please refrain from including details immediately after the subtitle; instead, place all information below it.

Please ensure that the font size and theme fonts are applied consistently throughout the entire document.

Please provide details on how the data were transitioned from bivariate to multivariate analysis, as well as how multicollinearity was assessed in the data analysis section.

Please correct the punctuation and grammar in the results sections.

Please verify the frequency and percentage in Table 1 against the total sample size. For example, the total for the age group category is 167, while the total frequency for the educational level is 230.

Additionally, the strengths of your study should be highlighted in the "Strengths and Limitations" section at the end of the discussion.

Please provide your resubmission along with a separate document containing a point-by-point response.

Reviewers' comments:

Reviewer's Responses to Questions

**Comments to the Author**

1. If the authors have adequately addressed your comments raised in a previous round of review and you feel that this manuscript is now acceptable for publication, you may indicate that here to bypass the “Comments to the Author” section, enter your conflict of interest statement in the “Confidential to Editor” section, and submit your "Accept" recommendation.

Reviewer #2: All comments have been addressed

2. Is the manuscript technically sound, and do the data support the conclusions?

Reviewer #2: Yes

3. Has the statistical analysis been performed appropriately and rigorously? 

Reviewer #2: Yes

4. Have the authors made all data underlying the findings in their manuscript fully available?

Reviewer #2: Yes

5. Is the manuscript presented in an intelligible fashion and written in standard English?

Reviewer #2: Yes

6. Review Comments to the Author

Reviewer #2: The authors have effectively addressed all of my comments, demonstrating a thorough understanding of the feedback provided. They have made significant revisions that enhance the clarity and depth of the manuscript, ensuring that it meets the necessary academic standards. Each point I raised has been carefully considered and incorporated into the revised version, which not only strengthens the overall argument but also improves the manuscript's coherence and flow. Given these substantial improvements and the authors' responsiveness to my suggestions, I am confident in recommending the acceptance of the manuscript. Their diligent efforts have resulted in a polished piece of work that contributes meaningfully to the field.

7. PLOS authors have the option to publish the peer review history of their article (what does this mean? ). If published, this will include your full peer review and any attached files.

**Do you want your identity to be public for this peer review?** For information about this choice, including consent withdrawal, please see our Privacy Policy .

Reviewer #2: **Yes: ** Teketel Ermias Geltore

---

## [Author Response · Author response to Decision Letter 3]

14 Oct 2025

Dear Academic Editor and Reviewers,

I sincerely appreciate your thoughtful feedback and constructive suggestions. Your input has greatly improved the quality of this manuscript and contributed to my growth as a researcher. Thank you very much for your valuable support.

I addressed each comment explicitly and attached it in a table as 'response to reviewers'.

---

## [Editor Report · Decision Letter 3]

2 Nov 2025

Assessment of awareness and attitudes of pregnant women toward labor epidural analgesia and factors influencing the decision to opt for it: A cross-sectional study.

PONE-D-24-60257R3

Dear Dr. Kassa,

We’re pleased to inform you that your manuscript has been judged scientifically suitable for publication and will be formally accepted for publication once it meets all outstanding technical requirements.

Kind regards,

Dereje Zewdu Assefa, BSc, MSc

Academic Editor

PLOS ONE
---

## [Editor Report · Acceptance letter]

PONE-D-24-60257R3

PLOS ONE

Dear Dr. Kassa,

I'm pleased to inform you that your manuscript has been deemed suitable for publication in PLOS ONE. Congratulations! Your manuscript is now being handed over to our production team.

Kind regards,

on behalf of

Professor Dereje Zewdu Assefa

Academic Editor

PLOS ONE